# Plant Elicitor Peptide (Pep) Signaling and Pathogen Defense in Tomato

**DOI:** 10.3390/plants12152856

**Published:** 2023-08-03

**Authors:** Alice K. Zelman, Gerald Alan Berkowitz

**Affiliations:** Department of Plant Science and Landscape Architecture, University of Connecticut, Storrs, CT 06269, USA; alice.zelman@uconn.edu

**Keywords:** pattern-associated molecular pattern, plant elicitor peptide, PAMP-triggered immunity, tomato defense signaling

## Abstract

Endogenous signaling compounds are intermediaries in signaling pathways that plants use to respond to the perception of harmful and beneficial organisms. The plant elicitor peptides (Peps) of plants are important endogenous signaling molecules that induce elements of defense responses such as hormone production, increased expression of defensive genes, the activation of phosphorelays, and the induction of cell secondary messenger synthesis. The processes by which Peps confer resistance to pathogenic microorganisms have been extensively studied in Arabidopsis but are less known in crop plants. Tomato and many other solanaceous plants have an endogenous signaling polypeptide, systemin, that is involved in the defense against herbivorous insects and necrotrophic pathogens. This paper explores the similarity of the effects and chemical properties of Pep and systemin in tomato. Additionally, the relationship of the Pep receptor and systemin receptors is explored, and the identification of a second tomato Pep receptor in the literature is called into question. We suggest future directions for research on Pep signaling in solanaceous crops during interactions with microbes.

## 1. Introduction

Current strategies for controlling crop disease rely on the exclusion of pathogens from fields and greenhouses, on cultural practices for reducing spread, on the destruction of diseased plant materials and infested soils, and on the application of chemicals and compounds that are designed to protect plants by attacking the attackers. The latter strategy can have serious repercussions for ecosystems and even human health; these compounds may be toxic or have other deleterious effects such as damaging non-target organisms in the ecosystem. In addition, the above practices do not prime the crops to defend themselves. A more effective management system could combine or modify existing practices with other methods that are designed to take advantage of the complex and effective detection systems that plants have gained through evolution for activating endogenous systems that combat pathogens [1]. Farmers could apply compounds that would activate the plant’s own early alert systems to activate the plant’s defense system before disease incursion into a crop area, tipping the plants’ priorities from growth towards defense. 

Plants must balance maximizing photosynthetic production to create material for growth, reproductive organ formation, and seed production; simultaneously, plants must minimize disease, predation, and loss of life. These factors are known as the so-called growth–defense trade-off [2,3]. Sustainable agriculture will increasingly rely on management of the growth–defense tradeoff [3]. In addition to passive defenses such as waxy coatings, callose deposition in the cell wall, and other physical barriers that passively reduce the success rate of biotic threats, the plant is always at the ready to respond to herbivores, pathogens, and infectious animals by ramping up the synthesis of defensive secondary metabolites [4]. The default homeostatic state is a watchful one: plants must successfully detect a threat and then produce a response to repel or reduce infection. Small molecules that activate infection-sensing watchdog receptors are termed pathogen-associated molecular patterns (PAMPs). PAMP-triggered immunity (PTI) in plants is analogous to basal immunity in vertebrate animals (as opposed to adaptive immunity) in that it is a generalized response to disease-causing microorganisms rather than against a specific single species of pathogen [5]. When such a threat is detected, proteins acting as watchdogs trigger a cascade of activities that include rapidly spreading the danger signal to other regions of the plant, which consequently increases the expression of genes whose products will protect the plant either directly or through the activation of other types of defenses. Some detector proteins seek evidence of pathogens, while others sense the presence of herbivores, for example, by detecting caterpillar oral secretions and frass [6,7,8].

One group of PAMP consists of microbe-associated molecular patterns (MAMPs), which are fragments of microbial molecules. The best studied MAMP is flagellin. Flagella are motile organs of many bacteria, and fragments of the flagellin protein are shed during bacterial invasion. Flg22 is a 22-amino acid moiety of flagellin that is capable of provoking an immune response from plants [9]. Flagellin (and flg22) is the ligand for FLAGELLIN-SENSING 2 (FLS2), a receptor in plant cell membranes. FLS2 is similar in structure to, for example, the Pep receptors (PEPRs), which bind the endogenous plant elicitor peptides (Peps) [10]. Plants also employ endogenous signaling compounds called damage-associated molecular patterns (DAMPs) that fine-tune and enhance the PTI response to harmful organisms [11]. DAMPs can be classified as either constitutive or inducible. Constitutive DAMPs play important homeostatic roles in normal, unstressed conditions, for example, in ATP and the components of cell walls; however, when fragments of the cell wall, ATP, or other constitutive DAMPs are detected in the apoplast, these molecules are immune-inducing signals. In contrast, inducible DAMPs (also called phytocytokines) are only active during stressed conditions and are considered as purely signaling molecules [12]. Peps and systemin are two types of DAMPs, and they both are considered phytocytokines, i.e., inducible DAMPs. Peps are short peptides that are post-translationally cleaved from precursor proteins called PROPEPs; systemin is also a short peptide that is post-translationally cleaved from its precursor, prosystemin (reviewed in [13]). Neither precursor protein has an N-terminal secretion signal [14]. In addition to being DAMPs, Peps and systemin are classified as peptide hormones. Table 1 lists the inducible plant DAMPs that are discussed in this review. Other inducible DAMPs not covered in this review include hydroxyproline-rich systemins (HypSys), which are found in solanaceous plants and, unlike Peps and systemin, have a N-terminal secretion signal [14]; and several peptides that are extensively studied in Arabidopsis, including rapid alkalization factors (RALFs) and phytosulfokines. 

Lori et al. [24] postulate that Pep sequences rapidly diverged and have no significant similarity across plant families, noting that there is “an astonishingly small sequence identity between PROPEPs”. This lack of similarity suggests that the Arabidopsis Peps (AtPeps) and other Peps cannot be assumed to act in identical ways, nor have identical properties; therefore, the study of Peps in other plants is not merely translating work in Arabidopsis to solanaceous crop plants. The lack of systemin signaling in non-solanaceous plants (and the functions that are performed by systemin that Peps perform in other plants) reinforces this idea. Tomato systemin was the first peptide hormone discovered [30]. It is activated after the detection of biotic threats, including herbivores and pathogens [31]. Pep signaling and systemin signaling therefore cooccur in tomato. The extent of overlapping or diverging function, as well as what components of their signaling pathways are shared, is currently not well understood. 

Broadly speaking, plant pathogens fall into two different categories based on their trophic strategies. The trophic strategy called necrotrophy involves a pathogen killing plant tissue and feeding on those dead cells. In contrast, biotrophs feed on living cells without causing tissue necrosis. Two commercially important tomato pathogens with a necrotrophic lifestyle are the oomycete *Pythium dissoticum*, which kills seedlings and roots, and the fungus *Botrytis cinerea*, the destructive causal agent for gray mold in seedlings, vegetative tissue, flowers, and fruits of hundreds of types of plant hosts. Another necrotrophic pathogen is the infamous causal agent of the late blight of potato, the oomycete *Phytophthora infestans*. *P. dissoticum*, was used by Trivilin et al. [32] in their study of tomato Pep (which they termed as SlPep6 due their identification of AtPep6 as its ortholog; hereafter, it is referred to as SlPep, as only one tomato Pep has been identified and subsequent works have referred to it as SlPep). Trivilin et al. [32] found that SlPep was able to reduce the severity of infection in tomato seedlings that were exposed to this fungus. Xu et al. [31] applied *B. cinerea* to tomato plants in their study of systemin and a putative tomato Pep receptor (PEPR), which they termed PEPR1/2 ortholog receptor-like kinase 1 (PORK1) and we henceforth call SlPEPR. The authors found that SlPEPR was needed for fully functional systemin signaling. Recently, it was also shown that SlPep reduces disease severity during *P. infestans* infection [25]. It should be noted that some pathogens exhibit biotrophic and necrotrophic behavior at different time points, for example, *Pseudomonas syringae* pv. *tomato* [33]. To date, it is unknown whether SlPep can reduce pathogen growth or disease severity during infection using a biotrophic or hemibiotrophic pathogen, and Pep efficacy in thwarting disease in other solanaceous crops such as eggplant, pepper, and potato is unknown. 

## 2. Biological Functions of Peps and Systemin

Post-translationally modified propeptides that lack a secretion signal (such as prosystemin and PROPEPs) appear to be an innovation of land plants, as they are absent in algae [34]. Systemin was first found to serve the plant as a warning signal and helped to defend against herbivorous insects that eat tomato plants [30]; it was subsequently shown to protect plants against necrotrophic microbes and was found in a specific clade. Systemin is restricted to the *Solanae* tribe of *Solanaceae* [35]. Other peptide hormones have since been found in other plants. Peps are a distinct family of peptide hormones that were first identified in *Arabidopsis* [16], and there are eight known Peps in *A. thaliana* [36]; maize Peps were the next to be identified [27]. 

Chemical, biophysical, and computational evidence shows that prosystemin is an intrinsically disordered protein, and the 18-residue C-terminal portion that comprises the mature signaling peptide is also disordered; the disordered nature of systemin is important to its activity [37]. It is unknown whether *SlPROPEP* and/or SlPep are disordered as well. 

Peps have been found experimentally in *Poaceae*, *Rosaceae*, *Solanaceae*, *Fabaceae*, and *Brassicaceae* [20,21,23,38]. In *Arabidopsis* and maize, Pep perception has been shown to lead to components of defense responses, such as hormone changes, calcium signals, phosphorylation, ROS (reactive oxygen species) generation, callose and lignin deposition, volatile compound production (VOC), accumulation of antimicrobial compounds, and the gene expression of, for example, protease inhibitors that inhibit herbivore growth [14,20]. 

Pep signaling has been linked with the defense against a variety of different organisms. Pep/PEPR signaling in *Arabidopsis* is induced by herbivores and has anti-herbivore defense effects [15]. AtPeps are also defensive against microbial pathogens including the hemibiotroph *Pseudomonas syringae* pv. *tomato* DC3000 and the necrotroph *Pythium irregulare* [16,17]. Soybean Peps are anti-nematicides [21]. In *Prunus persica*, Pep signaling after the application of any of several rosaceous Peps reduces bacterial disease symptoms [23]. Maize expresses ZmPep1, 2, and 3. ZmPep1 is protective against disease [27]; ZmPep3, on the other hand, activates anti-herbivore defenses [20]. Among solanaceous crops, eggplant, pepper, and potato emit volatile organic compounds in response to Pep application [20]. Broccoli PROPEPs’ expression was induced by the bacterial pathogen *P. syringae*; however, as of yet, the effects of broccoli Pep application on pathogenesis have not been reported, only its effects on salinity response [19]. Among the seven known rice Peps, OsPep3 was shown to have multifunctional defense effects in deterring both a piercing-sucking phloem-feeding herbivore (brown planthopper, *Nilaparvata lugens*) and pathogenesis by a fungal pathogen and bacterial pathogen (*Magnaporthe oryzae* and *Xanthomonas oryzae* pv. *Oryzae*) [22]. Solanaceous crops are affected by piercing-sucking insect pests, including aphids, psyllids, and whiteflies [39]; however, SlPep signaling and its relation to sap-sucking insect defense is unexplored in the current understanding. In tomato, compromising SlPep signaling increases the disease severity that is caused by a necrotrophic phytopathogenic fungus [32]. Moreover, SlPep can promote resistance against a necrotrophic phytopathogenic oomycete [25]. A broccoli Pep has been shown to be involved in salinity response and has also been implicated as having a role in development, particularly in root growth, where it has an inhibitory effect [19]. Interestingly, Pep signaling is a target of pathogen effectors. Two species of smut fungi produce Pep analogs that competitively inhibit Pep binding with its receptor, which suppresses host defense responses [40]. The peptide hormones including Pep and systemin evolve more quickly than the classical small organic molecules that were first identified as phytohormones, which is an advantage in the arms race between pathogen and plant [41]. 

Hormones have been found to control development and responses to conditions such as drought, flooding, heat, high salt or metal concentrations, wounding, and interactions with other organisms, including both beneficial and harmful microbes, plants, and animals [4,41,42]. Hormones or their secondary signals are active in both the local and distal organs of the plant. The downstream components of hormone signaling pathways can include the direct activation of enzymes that are based on hormone binding; secondary messenger signals such as calcium, ROS, and nitric oxide; and phosphorylation cascades such as MAP kinase activities that activate suites of transcription factors (reviewed in Zhao et al. [42]. 

Hormone molecules that are involved in plant responses to microbes include jasmonic acid (JA) and its precursor 12-oxo-phytodienoic acid (OPDA), salicylic acid (SA), abscisic acid (ABA), and ethylene (ET). JA (and other jasmonates) and SA are the two hormones that are classically associated with responses to the presence of phytopathogens and herbivorous insects. JA and OPDA are mainly involved in herbivore deterrence and defenses against infection by necrotrophic pathogens [15]. ET has roles in plant–microbial interactions, including responses to both beneficial and pathogenic bacteria, in addition to its important roles in growth, reproduction, and senescence. Lori et al. [24] synthesized variants of AtPep1 and ZmPep1 to contain the residues that are conserved within one *Nicotiana* Pep sequence (representing three species) and three *Solanum* Peps. This allowed wild-type tomato to emit ET in response to the altered *Arabidopsis* AtPep1 (AtPep1-SOL), but not the altered maize ZmPep1 (ZmPep1-SOL). The authors noted that Pep interactions with their receptor(s) must require additional residues beyond their currently identified functional motifs. SlPep was shown to increase JA and ET content [25]. All of the hormones mentioned can modulate other hormones’ signaling (“cross-talk”) [43]. SA was revealed to be antagonistic to JA (reducing JA levels and blocking JA signaling); JA can similarly reduce SA levels and block SA signaling [44]. 

An increasing body of evidence implicates ET in systemin and Pep signaling. In maize, Pep signaling increases JA and ET levels, which leads to protection against herbivores [20]. In the *Prunus* species, Peps were shown to increase the expression of ERF-1a and ERF-2b, which are involved in ET synthesis [23]. In tomato, the expression of the ET synthesis gene ERF1 was impaired when *SlPROPEP* transcript levels were reduced by viral-induced gene silencing, which suggested that ET signaling may be activated by SlPep [32]; a previous study [25] showed that exogenous SlPep application increased ET levels. Xu et al. [31] demonstrated that ERF-1b expression is also increased by systemin and that this change is actually increased in anti-*slpepr* RNA interference lines. Thus, Xu et al. [31] showed that reducing the expression of SlPEPR increased the expression of genes that encode ET biosynthesis enzymes. These two results seem to indicate that ET signaling is both hindered and increased by Pep signaling in tomato, depending on which area is examined in the SlPep pathway. Intriguingly, AtPep signaling through AtPEPR can, in addition to activating ET signaling, also compensate when some components in the ET pathway are disrupted [45]. In the same vein, it may be the case that SlPep signaling and systemin signaling are overlapping or redundant (at least in terms of pathogen defense responses), so that if one pathway is eliminated by a pathogen attack, the other can compensate and still defend the host plant. 

Hormones can modulate each other’s defense signaling pathways. For example, ET plays into both JA and SA signaling pathways. ET-responsive transcription factors can directly reduce SA biosynthesis and promote JA signaling [46]. JA and SA are often found to inhibit each other’s signaling pathway, and JA and ET are synergistic [43]; however, in PTI, the hormones JA, SA, and ET can all foster cooperative signaling [47]. When the SA level is elevated, the defense pathway engages against different types of pathogens and causes systemic acquired resistance (SAR); SA is associated with PAMP-triggered immunity (PTI) and defense responses against a wide variety of biotrophic and hemibiotrophic pathogens [48]. 

## 3. Local Responses to Peps

Calcium, reactive oxygen species (ROS), and MPK phosphorylation are signaling intermediaries that activate defense responses and are pertinent to Pep signaling. ROS move through the cell, interacting with molecules and damaging invading pathogen cell components, but also serve as a defense signal, such as in SA signaling and in Pep signaling that evokes immune responses [18,49]. Transient cytosolic calcium elevation (Ca^2+^ spikes) both induce ROS and are induced by ROS [49]. In *Arabidopsis*, ROS were shown to be activated by Pep-induced calcium signaling [18]; in tomato, AtPep1 caused an increase in ROS [50]. Also in tomato, ROS accumulation was impaired in *SlPROPEP*-silenced plants [25]. Interestingly, flg22-induced ROS production has been shown to decrease the fluidity of the plasma membrane in Arabidopsis and tobacco cells [51]. It is unknown what direct role this might play in plant defense; in any case, it would be intriguing to see if SlPep-induced ROS is also associated with plasma membrane rigidity. ROS are generally known to increase cell wall cross-linking, and the potential defensive advantages include increasing the difficulty of cell wall penetration for pathogens [52]. Flg22 and a wide range of other MAMPs and DAMPs increase ROS [53]. ROS production was not induced by systemin (compared with flg22 as a positive control) in an experiment reported by Xu et al. [31].

Like ROS, Ca^2+^ elevations are critical for plant defense responses among the host of pathways that require calcium signals [54]. Systemin and AtPeps cause calcium responses [55,56]. Ca^2+^ signals are some of the first responses to the perception of pathogens [56]. AtPep1 increases callose and lignin deposition, a late defense response that structurally protects plant tissues in a calcium-dependent manner [57]. The role of calcium signaling will therefore be important to study in the Pep signaling of solanaceous plants. A model of Pep signaling in tomato that incorporates both experimental findings and conjectural analogy to Pep signaling in *Arabidopsis* is shown in Figure 1. 

In addition to ROS production in response to Pep, peptide hormones cause defense-related gene expression changes. Systemin alters gene expression at the concentrations of fentomoles per gram of tissue [30,58]. Systemin-induced gene expression changes include the increased expression of allene oxidase synthase (AOS), lipoxygenase (LOX), calmodulin (CaM), proteinase inhibitors (PINs), and the prosystemin gene itself [56,58]. The first Pep discovered, AtPep1, increases defensin (PDF1.2) expression [16]. In maize, ZmPep3 could induce increased levels of AOS, PINs, and other herbivory defense-related transcripts [20]. As for SlPep, the accumulation of transcripts of the defense genes ACS, ERF1, LOXD, PR, and DEF2 was lower in *SlPROPEP*-silenced seedlings. Exogenous SlPep increased the expression of PR genes and, to a modest extent, WRKY33A, JA, and ET synthesis genes [25,32]. The interaction of MPK phosphorylation, cytosolic calcium signals, and ROS levels with Pep signaling in solanaceous crops is an important subject for future investigation. 

## 4. The Pep Receptor Identity Question

Though no secretory localization motifs have been identified within PROPEP or Pep sequences, it is known that Pep signaling can work through the perception of Pep binding as occurring from ligands to receptor proteins. Presumably, Peps act on plant cells as they are released from cells that are already under assault from pathogens. The Pep receptors AtPEPR1 and AtPEPR2 in *Arabidopsis* bind the eight AtPep peptides [17,28]. The AtPEPRs are members of a family of proteins called leucine-rich repeat receptor-like kinases (LRR-RLKs). LRR-RLKs are a prolifically duplicated protein family that appears to have originated in green algae [59] and that diversified early in the history of plant evolution [60]. LRR-RLKs serve as pattern recognition receptors, and their specialization is a plant-specific evolutionary trend that is crucial to pathogen defense in plants [61]. AtPEPRs have a leucine-rich repeat (LRR) region, a transmembrane (TM) domain, and a kinase domain. Within the kinase domain is a putative guanylyl cyclase (GC) domain that is linked to Ca^2+^ level elevation [56]. The portions of LRR-RLKs that contain LRRs serve as the ligand-binding domains of these proteins [61]. For example, the LRR regions of FLS2 bind flg22 in the extracellular portion of this receptor [62]. LRR domains in AtPEPR2 were found to stably bind AtPep1, and a crystal structure of AtPep1 that was bound to the LRR portion of AtPEPR2 was solved [29]. Accordingly, the specificity of Pep binding is likely to be located to the extracellular LRR-containing regions in the Pep receptors of other species in addition to *Arabidopsis*. AtPep1 was shown to be in an extended structure when bound to AtPEPR1 [29]. Chemical, biophysical, and computational evidence has shown that prosystemin is an intrinsically disordered protein, and the 18-residue C-terminal portion that comprises the mature signaling peptide is also disordered; the disordered nature of systemin is important to its activity [37]. 

Identifying SlPEPR is a key step in understanding Pep signaling in solanaceous crops. SlPep-SlPEPR binding has not yet been demonstrated based on pull-down assays or other physical binding studies, but there are two proposed SlPEPRs in the literature. Several publications [24,31,63] have all identified the same protein sequence as a likely PEPR in tomato due to it having the most similarity to AtPEPR1/AtPEPR2 (a transcriptome profiling study reported that the expression of the gene encoding this protein is upregulated in drought-tolerant varieties of tomato during drought stress but is not upregulated in susceptible cultivars during drought stress [64]. In that study, SlPEPR is termed receptor-like protein kinase INRPK1c). A predicted SlPEPR has been implicated in the signaling pathways of both Peps [24] and systemin [31]. It has been demonstrated by Xu et al. [31] that fully active systemin signaling requires this tomato ortholog for the AtPEPRs. Xu et al. [31] investigated SlPEPR in the context of systemin’s ability to reduce herbivory and necrotrophic fungal infection. An initial draft of Xu et al.’s paper conjectured that the then unidentified systemin receptor might be the tomato PEPR, but during peer review, it was shown by another research group that their SlPEPR candidate was not the systemin receptor. In fact, SlBRI1 was once thought to be the systemin receptor based on biochemical evidence; however, this was subsequently shown to be incorrect, and SlBRI1 was instead identified as the brassinosteroid receptor [65]. 

SYR1 was shown to be a bona fide receptor that binds systemin; SYR1 and SYR2 mediate systemin signaling [26] in tomato. Xu et al. [31] predicted that SYR1 and SYR2 are the two proteins with the greatest similarity to SlPEPR from among the range of tomato LRR-RLKs that they analyzed, which raises interesting questions about the evolutionary relationship between systemin and Peps. Table 2 presents a summary of the proposed Pep and systemin receptors, along with sequence identifiers for the National Center for Biotechnology Information (https://www.ncbi.nlm.nih.gov/, accessed on 30 May 2023) database and the solanaceous database, Solgenomics (https://solgenomics.net/; accessed on 2 January 2022) [66] (it should be noted that the NCBI sequence names for these proteins are frequently non-specific and in one case inaccurate, as noted in Table 2). Systemin sequences differ from the Peps significantly and are not considered to be homologous to the Pep family in the current understanding. It is certainly curious that systemin and Pep receptors are so similar when the ligands are not detectably homologous; however, this situation is not without precedent. The JA-OPDA receptor Coronatine Insensitive 1 (COI1) and the auxin receptor Transport Inhibitor Response 1 (TIR1) are considered by some to be homologous [67]. Another possibility is that systemin is actually a Pep that rapidly evolved and no longer has sufficient sequence similarity to detect homology with the Pep family. This is not impossible given the very low conservation of Peps in different plant families. Wang et al. [26] provided a tree that includes SlPEPR, the AtPEPRs, and SYR1 and SYR2; however, their tree does not include other *S. lycopersicum* LRR-RLKs, so this tree does not offer additional insight into the relationship between the PEPRs and SYR1 and SYR2. Lori et al. [24] did not include any other tomato proteins in their phylogenetic analysis. In contrast, Rahman et al. [63] predicted that there are actually two SlPEPRs in their study of tomato guanylyl cyclase proteins. Peps and their cognate receptors, PEPRs, have been of interest in the context of cyclic nucleotide signaling. Cyclic GMP (cGMP) has been investigated as a signaling compound that, among other effects, activates the opening of calcium channels in Pep signaling [18,56]. The protein that was identified as a putative SlPEPR by Lori et al. [24] and Xu et al. [31] was investigated by Rahman et al. [63], who named it guanylyl cyclase 18 (SlGC18). Interestingly, SlPEPR was predicted by Rahman et al. [63] to be most similar to a different putative GC, which they termed SlGC17, and the authors asserted that both of these two proteins are likely SlPEPR proteins. The discrepancy between published phylogenetic trees suggests that it is worthwhile to conduct a phylogenetic analysis of PEPRs in Solanaceae, as the different trees have different implications for the origins and mechanisms of Pep signaling. Perhaps these two LRR-RLKs physically interact to transduce Pep and systemin signaling, or perhaps both may be required to interact with some third signaling component. AtPEPR, BRASSINOSTEROID-INSENSITIVE 1 (BRI1), and the flagellin receptor FLS2 bind with the somatic embryogenesis receptor kinase (SERK) isoform BRI1-ASSOCIATED KINASE 1 (BAK1) in Arabidopsis (reviewed in DeFalco and Zipfel, [68]). SlSERKs may play a similar role in tomato by interacting with SlPEPR, SlSYR1/2, and/or SlGC17, and this warrants future study. Pulldown assays to determine whether there is physical interaction between SlPep and SlGC17, SlPep and SlPEPR, and SlPEPR and SlGC17 would be informative as well.

## 5. The Agricultural Context 

Use of the bacterial harpin protein was the first example of the utilization of the strategy of priming plant immunity as a commercial disease prevention strategy [70]. Harpin is an example of a microbe-associated molecular pattern (MAMP) that elicits HR, which causes ETI [71]. Harpin is available for commercial use, but its cost may be impractical for farmers. While it was recently shown to effectively reduce the growth of the pathogen *Pythium ephanidermatum* on cannabis [72], harpin was ineffective as a biocontrol in several studies, such as on the bacterial canker of tomato [73] and bacterial spot of tomato [74]. DAMP peptides present an alternative [1]. The endogenous DAMPs are active at lower concentrations than MAMPs; they are also less expensive to synthesize than (much larger) full-length proteins such as harpin (up to ~500 mer [75]). Because a smaller concentration would need to be applied for the same effect, and because the peptide has far fewer amino acids than MAMP proteins such as harpin and thus would be less expensive, this raises the possibility that DAMPs could be an efficient treatment for crops [76]. Additionally, being an endogenous plant compound that is easily broken down, peptide DAMPs would not persist in the environment in a manner that many pesticides do. Therefore, commercial solutions that use synthesized versions of endogenous plant signals to prepare plants to deal with a future disease might be significantly cheaper and have few to no environmental repercussions compared with current strategies [76,77]. An approach that combines systemin treatment with beneficial microorganism application has shown promise [78]. SlPep and other solanaceous Peps may therefore have a future role in crop protection against pathogenic microorganisms. 

## Figures and Tables

**Figure 1 plants-12-02856-f001:**
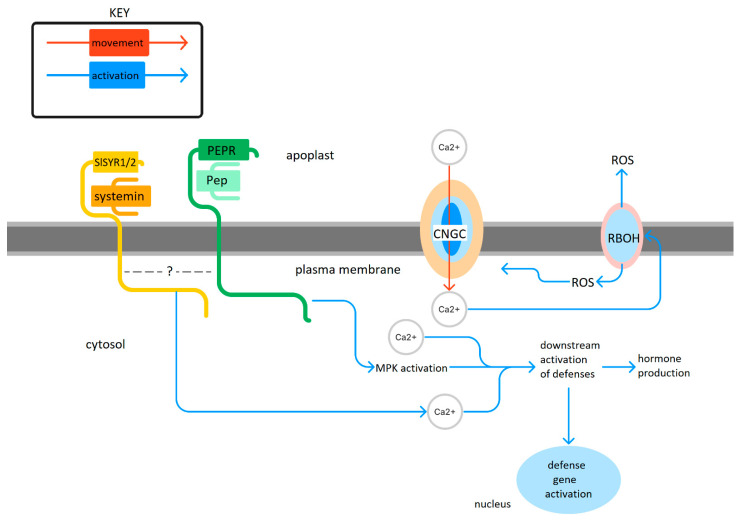
A speculative model of Pep signaling based on the evidence from signal transduction in Arabidopsis. There is evidence that SlPep causes ROS accumulation. If the signaling cascade is similar to that of Arabidopsis, these ROS are in large part produced by respiratory burst oxidase homolog (RBOH), and ROS activates a calcium channel(s) such as the CNGCs. Calcium signaling activates more accumulation of ROS in a feedback loop. PEPR’s kinase domain may result in MPK activation, which can lead to the downstream activation of defenses such as hormone responses and defense gene expression. Systemin signaling interacts with Pep signaling components, perhaps due to the binding of its receptor to PEPR or through other means. Systemin causes cytosolic calcium increases, which likewise leads to downstream defense responses.

**Table 1 plants-12-02856-t001:** A list of the inducible DAMPs discussed that have been experimentally verified, including the eight Arabidopsis Peps and Peps from broccoli, tomato, soybean, peach, rice, and maize; and systemin.

DAMP	Species	Receptor (Predicted) *	Demonstrated Biological Activity of Pep Signaling	References
AtPeps 1-8	Arabidopsis	AtPEPR1, AtPEPR2	anti-chewing insect herbivore; anti-hemibiotrophic pathogen; anti-necrotrophic pathogen; root development	[15,16,17,18]
BoPep4	broccoli	unreported	salinity response	[19]
GmPep1, GmPep2, GmPep3	soybean	GmPEPR1a, GmPEPR2a, GmPEPR2	anti-chewing insect herbivore, anti-nematode	[20,21]
OsPep3	rice	OsPEPR1a, OsPEPR1b	anti-bacterial, anti-fungal; anti-piercing-sucking insect herbivore	[22]
PpPep1 and 2	peach	PpPEPR1a, PpPEPR1b	anti-bacterial necrotrophic pathogen	[23]
SlPep	tomato	SlPEPR, SlGC17	anti-necrotrophic pathogen	[24,25]
SlSystemin	tomato (homologs in Solanae clade)	SlSYR1, SlSYR2	anti-necrotrophic pathogen; anti-chewing insect herbivore	[26]
ZmPep1, ZmPep3	maize	ZmPEPR1a, ZmPEPR2a	anti-chewing insect herbivore; anti-necrotrophic pathogen	[20,27]

* There is biochemical (binding) assay experimental confirmation that AtPEPR1 and AtPEPR2 are Pep receptors for Arabidopsis Peps [17,28,29] and that SlSYR1 is a systemin receptor [26]. Abbreviations: DAMPs, damage-associated molecular patterns; Pep, pathogen elicitor peptide; PEPR, Pep receptor; GC, guanylyl cyclase; SYR, systemin receptor. Plant scientific names: Arabidopsis, *Arabidopsis thaliana*; broccoli, *Brassica oleracea* var. *italica*; maize, *Zea mays*; peach, *Prunus persica*; rice, *Oryzae sativa*; soybean, *Glycine max*; tomato, *Solanum lycopersicum*.

**Table 2 plants-12-02856-t002:** Proposed Pep and systemin receptors and their biological roles. If a publication referred to a receptor with a different name than is used in this work, it is noted in the “Alternate Name” column.

Protein	Alternate Name	Biological Role	Reference	SolGenomics ID	NCBI Identifier
SlPEPR	SlPORK1	proposed Pep receptor	[24]	Solyc03g123860	XP_004235511.1
	SlPEPR	systemin signaling pathway component	[31]		
	INRPK1c	expression upregulated in drought-susceptible cultivars during drought stress	[64]		
	SlGC18	proposed PEPR; guanylyl cyclase	[63]		
SlGC17	n/a	proposed PEPR; guanylyl cyclase	[63]	Solyc03g112580.2.1	XP_004236236.1
SlSYR1	*	systemin receptor (biochemical evidence)	[26]	Solyc03g082470.2.1	XP_004235118.1
SlSYR2	n/a	systemin receptor	[26]	Solyc03g082450.2.1	XP_004235119.1
SlBRI1	SR160; cu3 receptor	proposed systemin receptor, later shown to be incorrect by Malinowski et al., 67	[69]	Solyc04g051510.1.1	NP_001296180.1

* Incorrectly named “leucine-rich repeat receptor-like protein kinase PEPR1” in NCBI protein database. Abbreviations: NCBI, National Center for Biotechnology Information; Pep, pathogen elicitor peptide; PEPR, Pep receptor; PORK1, PEPR1 ortholog receptor-like kinase; GC, guanylyl cyclase; SYR, systemin receptor; BRI1, brassinosteroid insensitive 1.

## Data Availability

No new data were created for this manuscript.

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
