# Peer review of "Plant Elicitor Peptide (Pep) Signaling and Pathogen Defense in Tomato"

_plants, 2023, doi:10.3390/plants12152856_

Round 1
Reviewer 1 Report
see attachment.

Reviewer 2 Report
Peps and systemin are widely distributed among angiosperms plant species and play important roles in multiple aspects of plant growth and defense. This article reviews Peps and systemin related research from three main aspects which are basically forward-looking and of great significance. However, there are still some deficiencies in this paper, such as lack of focus and clarity. The followings are relevant suggestions.
1. Introduction. Make a list of DAMP species of different plants (Arabidopsis thaliana, tomato and other crop)
2. Make a list to introduce the identification、receptor(perception) and function (response) of Peps and systemin in different plants.
3. It is recommended adding the 'biological function of Peps and systemin' and merge 'Local responses to Peps' into this section.
4. In addition to sequence similarity, it is recommended to introduce the relationship between Peps and systemin from biosynthesis, signal transduction and response.
The sentences are smooth, but some are not rigorous enough.
Round 2
Reviewer 1 Report
The authors has made considerable effort to improve the manuscript and it is acceptable for publication in PLANTS.